# Strain variation and anomalous climate synergistically influence cholera pandemics

**Xavier Rodó** [1,2☯] *, **Menno Jan Bouma** [2☯], **Miquel-Àngel Rodríguez-Arias** [3], **Manojit Roy** [4], **Pau De Yebra** [5,6], **Desislava Petrova** [2], **Markel García-Díez** [7], **Mercedes Pascual** [8]

**1** ICREA, Barcelona, Spain, **2** CLIMA (Climate & Health) Group, ISGlobal, Barcelona, Spain, **3** CMIMA, CSIC, Passeig Marítim, Barcelona, Spain, **4** Department of Ecology and Evolutionary Biology, University of Michigan, Ann Arbor, Michigan, United States of America, **5** Department of Experimental Limnology, Leibniz Institute of Freshwater Ecology and Inland Fisheries, Stechlin, Germany, **6** Department of Wildlife Diseases, Leibniz Institute for Zoo and Wildlife Research, Berlin, Germany, **7** Predictia Intelligent Data Solutions S.L., Santander, Spain, **8** Department of Biology and Department of Environmental Sciences, New York University, New York City, New York, United States of America

☯ These authors contributed equally to this work.

\* xavier.rodo@isglobal.org

## Abstract

### Background

Explanations for the genesis and propagation of cholera pandemics since 1817 have remained elusive. Evolutionary pathogen change is presumed to have been a dominant factor behind the 7th "*El Tor*" pandemic, but little is known to support this hypothesis for preceding pandemics. The role of anomalous climate in facilitating strain replacements has never been assessed. The question is of relevance to guide the understanding of infectious disease emergence today and in the context of climate change.

### Methodology/principal findings

We investigate the roles of climate and putative strain variation for the 6th cholera pandemic (1899–1923) using newly assembled historical records for climate variables and cholera deaths in provinces of former British India. We compare this historical pandemic with the 7th (*El Tor*) one and with the temporary emergence of the *O139* strain in Bangladesh and globally. With statistical methods for nonlinear time series analysis, we examine the regional synchrony of outbreaks and associations of the disease with regional temperature and rainfall, and with the El Niño Southern Oscillation (ENSO). To establish future expectations and evaluate climate anomalies accompanying historical strain replacements, climate projections are generated with multi-model climate simulations for different 50-year periods.

The 6th cholera pandemic featured the striking synchronisation of cholera outbreaks over Bengal during the El Niño event of 1904–07, following the invasion of the Bombay Presidency with a delay of a few years. Accompanying anomalous weather conditions are similar to those related to ENSO during strain replacements and pandemic expansions into Africa and South America in the late 20th century. Rainfall anomalies of 1904–05 at the beginning of the large cholera anomaly fall in the 99th percentile of simulated changes for the regional climate.

**Data Availability Statement:** Cholera datasets are publicly available at: https://github.com/xrodo6/Cholera-Bangladesh.

**Funding:** We thank the Indian Institute for Tropical Meteorology at Pune (http://www.tropmet.res.in) for supplying meteorological data and the early support for our cholera work by NSF-NIH (Ecology of Infectious Diseases Program, NSF 0545276 and 0430120) and NOAA (Oceans and Health, NA040AR460019) to M.P. X.R acknowledges the support of the PERIS-PICAT project of the Catalan Dep. Salut and PARA-CLIM-CHANDIRGARGH of the New Indigo EU-India program. We acknowledge support from the Spanish Ministry of Science and Innovation through the "Centro de Excelencia Severo Ochoa 2019–2023" Program (CEX2018-000806-S), and support from the Generalitat de Catalunya through the CERCA Program. The funders had no role in study design, data collection and analysis, decision to publish, or preparation of the manuscript.

**Competing interests:** The authors have declared that no competing interests exist.

## Conclusions/significance

Evolutionary pathogen change can act synergistically with climatic conditions in the emergence and propagation of cholera strains. Increased climate variability and extremes under global warming provide windows of opportunity for emerging pathogens.

## Author summary

Climate anomalies appear to have played an important role in facilitating the establishment of a new invasive strain in the 6th cholera pandemic, with climate and environmental conditions similar to those underlying strain changes associated with ENSO in today's Bangladesh. The evolutionary change of pathogens can act synergistically with climatic conditions in the replacement and geographical propagation of emerging strains. Increased climate variability and extremes under global warming would thus provide new windows of opportunity for emerging pathogens/re-emerging cholera outbreaks.

## Introduction

The global spread of potentially fatal infectious diseases, such as plague, flu and cholera, had a significant impact on human history through large demographic and economic consequences. Although the clinical description of cholera is recorded in classical publications, and the disease has long been endemic in the Indian subcontinent, pandemic excursions of cholera from its endemic regions were only first reported in the 1820s. In the 19th century cholera caused social shock waves reminiscent of those of plague in earlier centuries, striking regardless of age, sex and social class. Some strains of toxigenic *Vibrio cholera* resulted in explosive outbreaks when introduced into immunologically naive populations with poor sanitary infrastructure, as was evident in the devastating 2010 cholera epidemic in Haiti following the earthquake disaster [1–3]. Since the introduction of major and widespread hygienic initiatives for waste disposal and water supply during the 20th century, cholera has become a disease of poverty [4], with an estimated 2.9 million cases and 95,000 deaths in 69 endemic countries each year [5]. The number of cases has risen sharply in many countries since 2022 with a global resurgence of the disease that is still on-going [6].

The origin of cholera and in particular the role of the environment has been debated for the last two centuries, long after the discovery by Koch in 1883 of its causative agent, the bacillus *Vibrio cholerae* [7]. On the one hand, the worldwide expansion of both the 6th and the 7th pandemics has occurred in the form of several waves [8], with the successive colonisation of the different continents, allegedly following favourable climate and socioeconomic conditions in crowded low-income countries in Asia, Africa and ultimately, the Americas [9]. Thus, climate conditions would have facilitated the regional spread of the disease once seeded into a new area by human mobility. Human mobility in the context of climate variability has also been documented recently within the *El Tor* biotype for 1991 isolates from several countries (e.g. China [10]; Thailand [11]; and Angola [12]).

On the other hand, genetic changes of the pathogen associated with severe outbreaks and clinical and epidemiological manifestations have implicated evolutionary change in *V. cholerae* as responsible for previous pandemics [13–15]. Travel can fuel propagation of the bacterium under either scenario. Evidence of travel from cholera-affected cities as a source of the pathogen was presented for the large Danish 1853 cholera outbreak which killed 3.4%–8.9% of the

population, with the highest mortality among seniors (16%) and the lowest among children (2.7%) [16].

The 7th pandemic is specifically associated with the propagation and partial replacement in endemic parts of India and Bangladesh of the *Classical* biotype by the *El Tor* strain in the 1960s. Multiple introductions in Africa from its putative origins in the Bay of Bengal have also re-emphasised in recent years the importance of human mobility to pandemic cholera. The more recent appearance of the *O139* strain temporarily raised the prospect of a new cholera pandemic [17]. The isolation and typing of cholera only possible since the 1980s led to the classification of the biotype preceding *El Tor* as *Classical*, which would have been responsible for the 5th and 6th pandemics and most likely for the preceding ones [18]. The extensive genetic variation within the *Classical* biotype is likely to have fostered the 6th pandemic. The exact origins and impact of the 1899–1923 pandemic in former British India remained however rather obscure due to over 800,000 cholera deaths in 1900, and the Viceroy's moratorium on related publications for fear of generating adverse publicity for the colonial enterprise.

To date, the role of climate conditions and strain variation has not been thoroughly investigated as a driver of pandemic cholera. Increasingly available genomic and environmental data enable consideration of these factors in ways previously impossible. There is also growing interest in the potential contributions of both climate conditions and evolutionary change to the emergence of other pathogens, including for example chikungunya, dengue and more recently Zika [19,20]. In this study, we analyse extensive data from historical records on cholera mortality covering the onset of the 6th and 7th cholera pandemics, and present evidence supporting the relevance of both strain novelty and anomalous weather conditions. We end with potential implications for (pandemic) disease emergence and variability in a warmer world.

## Data and methods

**Cholera data.** The monthly cholera mortality figures (1893–1939) and percentages of villages with reported cholera deaths were collated from British libraries based on the annual reports of the sanitary commissioners for districts in Bengal [21], Punjab [22] and Calcutta [23]. In addition to records for the non-endemic Bombay Presidency, these reports provide spatially explicit data for the other endemic Provinces traversed by the Ganges and Brahmaputra the 24 Bengal districts described in previous studies [24] (see also the S1 Supplementary Information). Districts in Orissa and Bihar which were part of the Bengal Presidency before 1912, were excluded from the analysis. In addition, incomplete reports are available for case fatality rates of "boats" men living in the harbour (e.g. "the native floating population") in Calcutta (Fig 1, 1900–1912), and for cholera mortality in the Punjab by age group from 1908 to 1919 (S1A Fig).

## Climate data

For the period of 1893 to 1939, climate data were obtained from different sources due to the lack of available products covering both the 19th and the 20th centuries together. Rainfall and minimum and maximum temperature from surface station data were retrieved from the historical archives of former British India [25] (Fig 1; see also the S1 Supplementary Information). For SSTa in the historical period, we used the Kaplan Extended SST V2 [26,27]. This data set includes gridded SST anomalies from 1856 from 1856 to THE present and is based on a climatological period between 1951 and 1980. We also generated averages from land station values for both monthly temperature in Dacca (˚C) and the North East India Rainfall (NEIR, mm) from the climatic database of the Indian Institute for Tropical Meteorology at Pune (http://

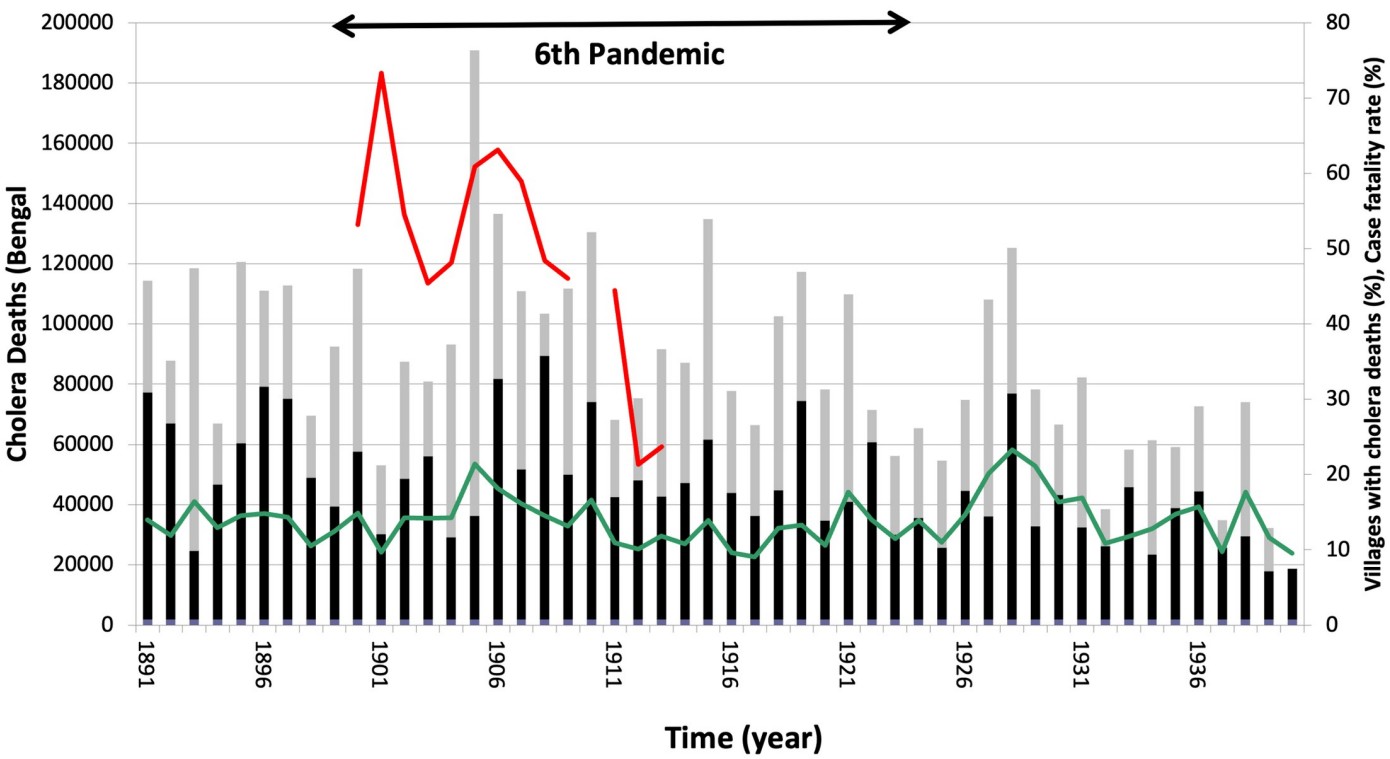

**Fig 1. Cholera deaths in former Bengal Province.** Deaths between February-July (black bars) and between August-January (grey bars). Case fatality rate (%) in Calcutta harbour, the "floating population" (Red) and the % of villages in Bengal with cholera deaths (green) on secondary Y axis. Data corresponds to the 24 districts of Bengal, with 86,353 villages (in % of villages with cholera deaths).

www.tropmet.res.in). In addition, to represent rainfall in more recent Bangladesh spatially, we used the gridded rainfall data from Global Precipitation Climatology Project (GPCC, the Full Data Reanalysis (FD) V6) [28]. For rainfall time series, NEIR regional precipitation was preferred to Dacca's local rainfall as it represents a better way to characterise the total water accumulation in the catchment area of interest (Figs 2–4).

The Niño3.4 SST index of ENSO over coastal South America was considered to address associations of cholera with this major driver of global climate variability (Fig 5) (http://www.esrl.noaa.gov/psd/people/cathy.smith/best/).

## Statistical analyses

The synchrony of cholera mortality in the different districts was evaluated with Cross-Wavelet power spectra (Fig 2A), computed from the individual wavelet power spectra for the two individual time series (WPS) [29–31]. This approach was selected to specifically examine associations that are transient in time and to analyse time series that are themselves non-stationary. Under such conditions, the classical Fourier power spectrum can be problematic since it averages over time the contributions of the different frequencies, and therefore these contributions cannot be localized in time. In contrast, the Wavelet spectrum provides a decomposition of the variance of a time series into different frequencies explicitly over time. Similarly, the resulting Cross-Wavelet power spectrum indicates not just whether, but when in time, two time series exhibit variance at similar frequencies. Cross-spectra were calculated for pairs of selected districts across a large geographical region spanning from the coast to

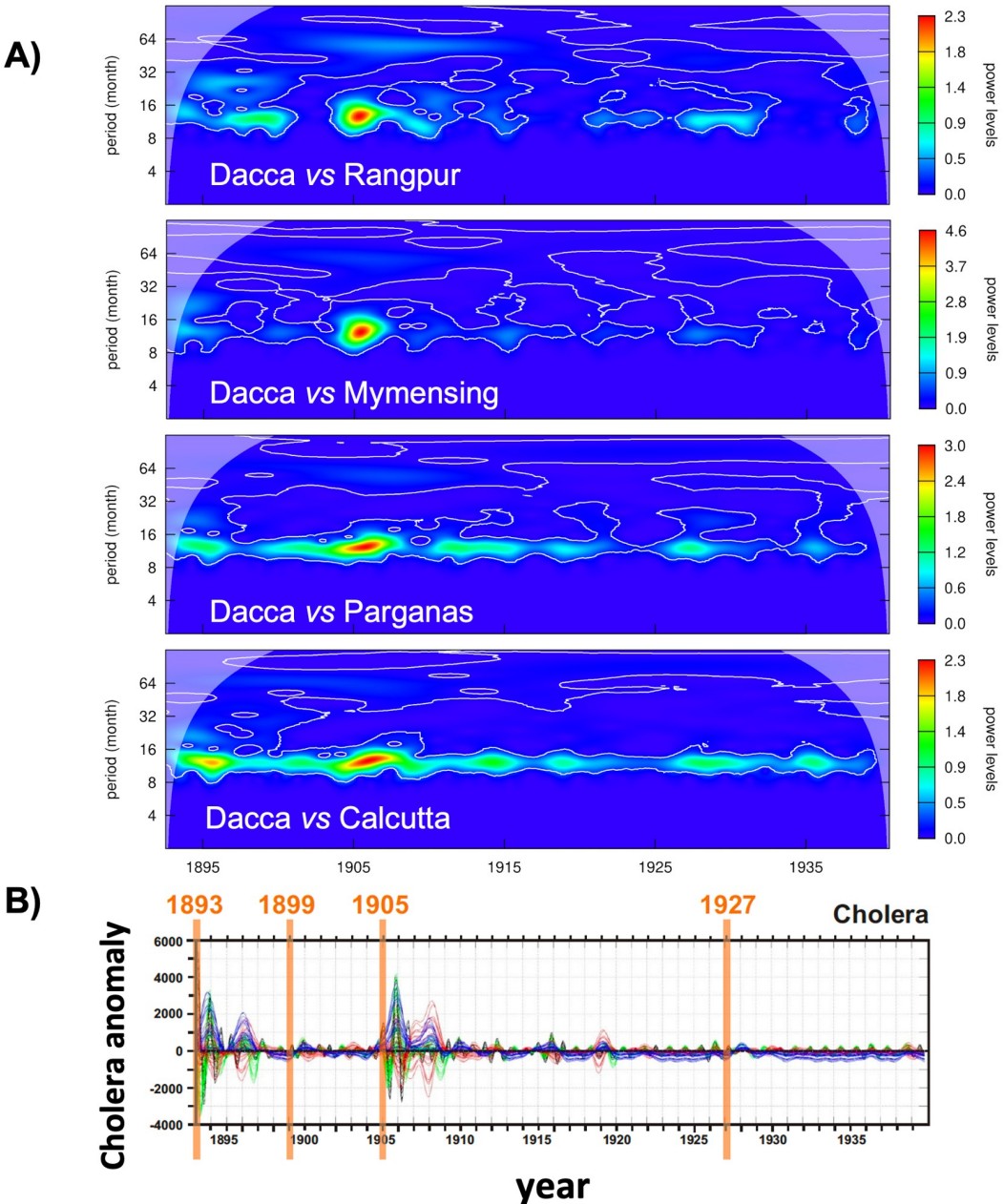

**Fig 2.** A) Wavelet cross-spectrum power analysis between monthly cholera mortality in Dacca and that of selected districts in the different regions of former Bengal (Rangpur, Mymensing, Parganas and Calcutta) from 1893 to 1935. Each plot shows the intensity of the cross-spectrum as a function of time (in the x axis) and period (in the y axis). Areas in the plots with high power indicate synchronization. Those where the signal is significant are surrounded by black lines. Areas in the plots with high power at the same time indicate synchronization, namely for the years 1904–07. These were. B) Seasonal-to-interannual components of cholera for all districts in Dacca for the interval 1893–1940, reconstructed with Singular spectrum analysis (MCSSA). Note the two times (1893 and 1905) around which the variability of cholera anomalies at all temporal scales is more pronounced, denoting an intense alteration of the normal pattern (see S1 Supplementary Information *methods for description*). Note also the lack of spectral response around 1927.

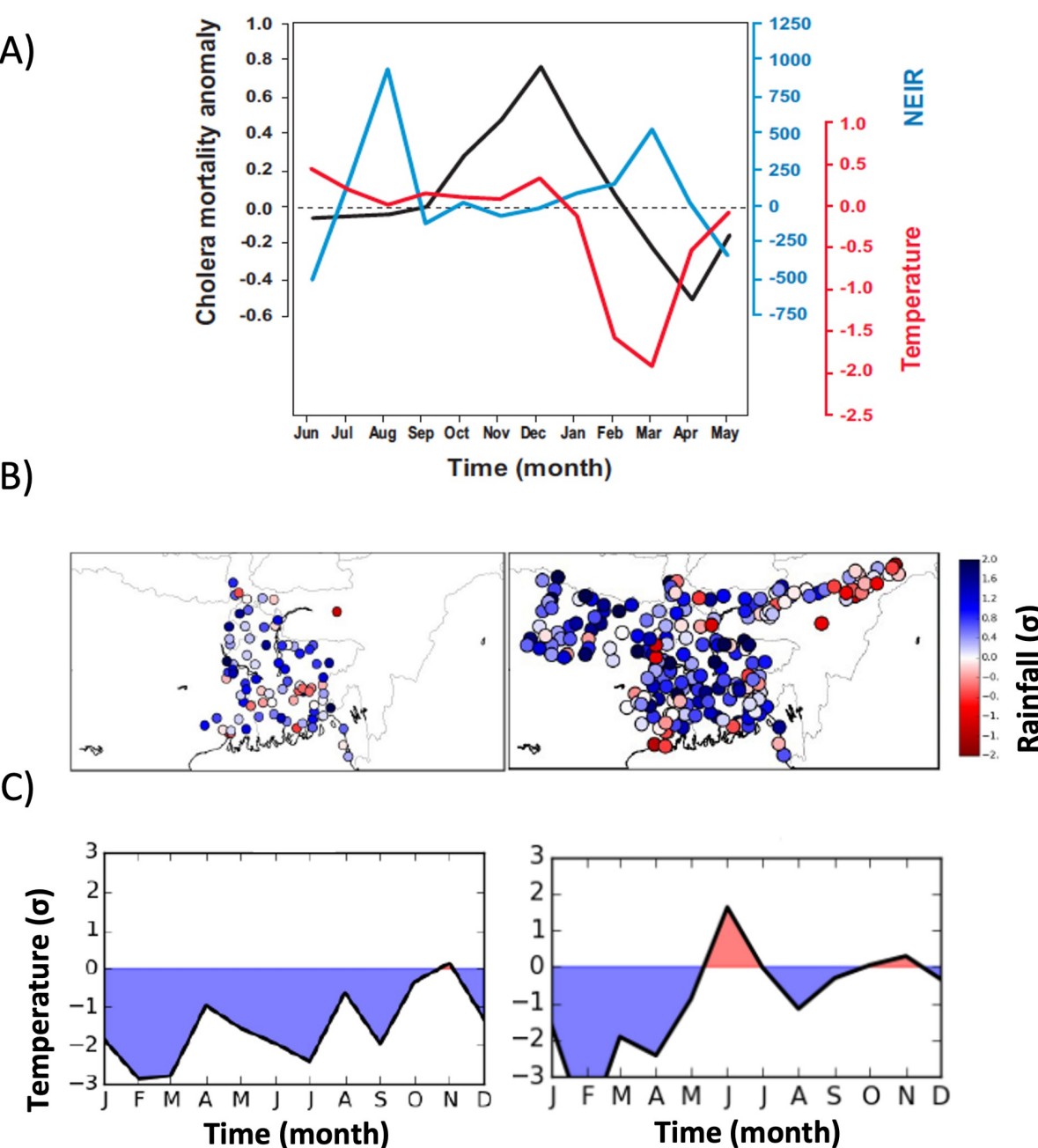

**Fig 3.** A) Difference between the monthly average of the 1904–07 event and the long-term (1893–1939) seasonal cycles of monthly cholera mortality anomaly (black), rainfall from NEIR (blue) and temperature (red) in Dacca (*note that the interval 1904–07 was not used for the derivation of the seasonal cycle*). B) Rainfall anomalies for Bengal in 1893 (left) and Bengal, Bihar and Assam in 1905 (right). Colours denote the number of standard deviations from the mean of rainfall in the intervals 1893–1939 (B, left); and 1901–2010 (B, right). Precipitation anomalies are shown for June-July-August-September (JJAS). C panels display temperature anomalies (deg. ˚C) as standard deviations from means in the respective interval years, as in panel 3B.

the North of former Bengal along the Brahmaputra basin, with all pairs including the district of Dacca as a central reference (Fig 2). The R-package WaveletComp1.1 was used for this purpose. (For detailed description of these methods and application to epidemiology, see [32]).

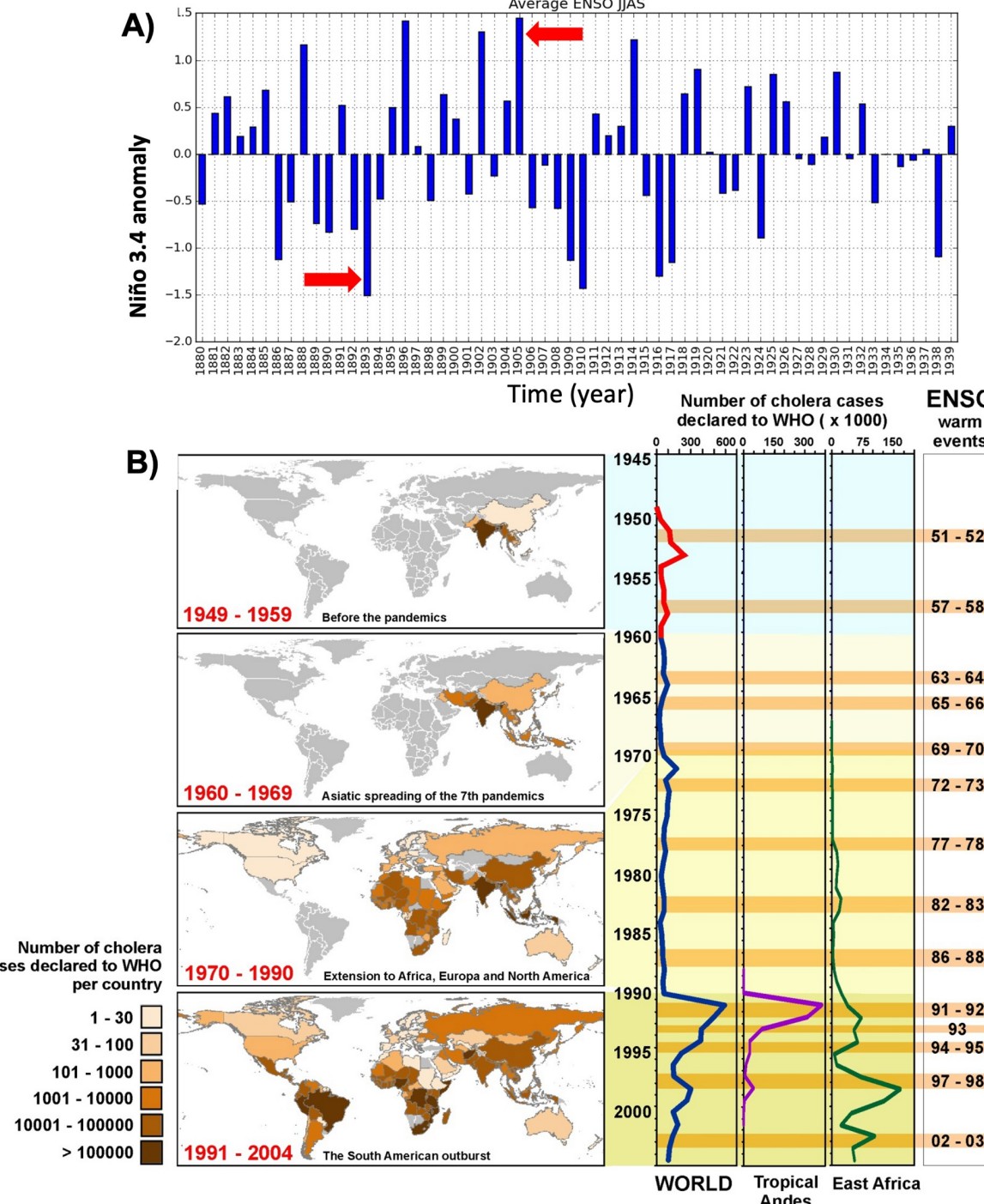

**Fig 4.** A) ENSO index data from 1880 to 1939 retrieved from http://www.esrl.noaa.gov/psd/people/cathy.smith/best/. Arrows indicate locations of the 1893 and 1905 strong ENSO events. B) Left: Global propagation of cholera in the 7th pandemic (1950–2004) showing colonisation of different continents in terms of cholera cases reported by country to WHO. Right: Number of accumulated cholera cases (x1000) in the world, tropical Andes and East Africa, and the successive El Niño events on record.

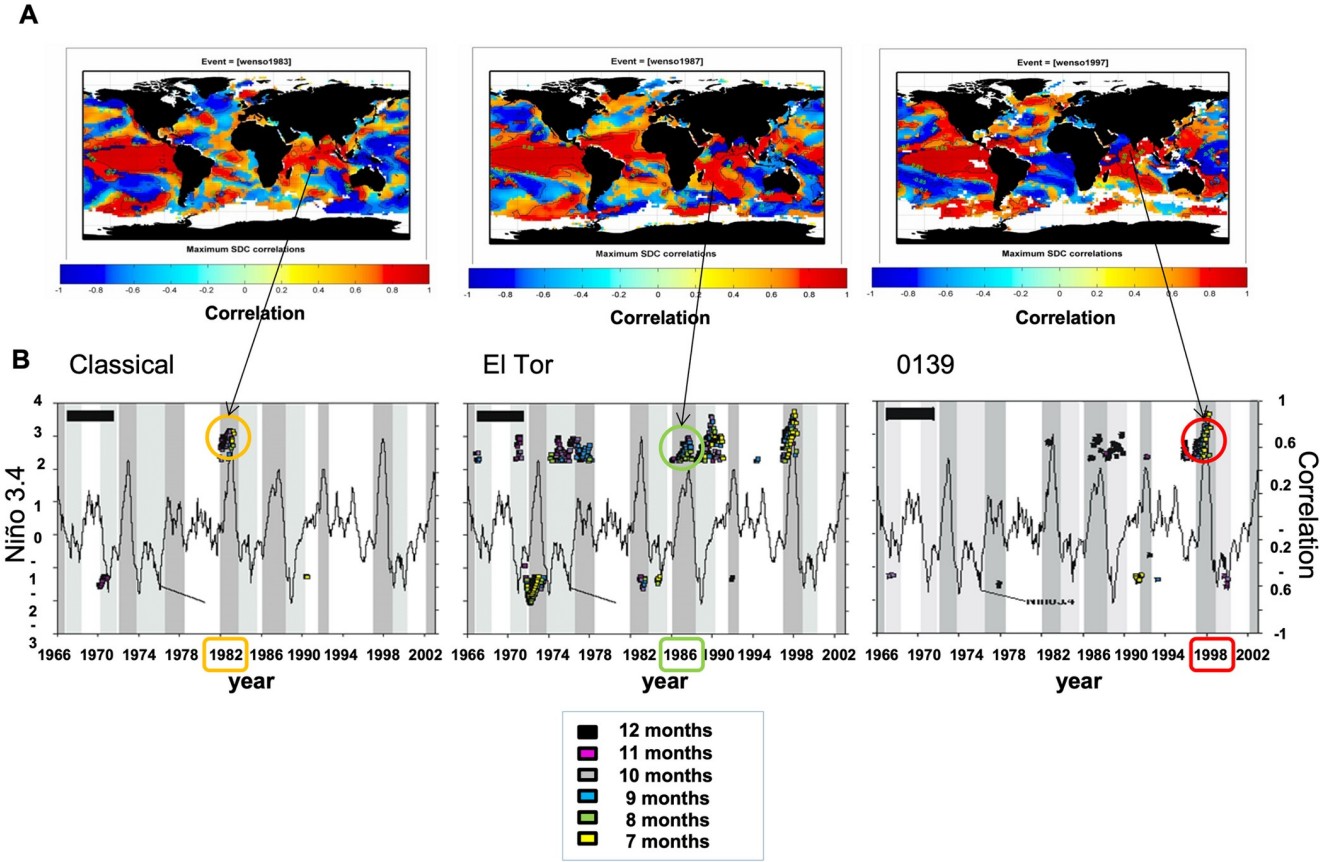

**Fig 5.** A) Scale-dependent correlation (SDC) [29] analysis maps between sea-surface temperatures (SST) anomalies and ENSO (Niño 3.4). B) SDC analysis between total cholera cases for each of the three different strains in Bangladesh (*Classical*, *El Tor* and *O139*), and El Niño (*Niño3.4 index*), with a lead time of the climate variable of 7 to 12 months. Depicted spatial correlations correspond to the correlation maximum attained at the times when epidemics for each strain occurred in Bangladesh and the spatial replacement is known to have occurred (1982 for *Classical*, 1986–88 for *El Tor* and 1995–97 for *O139*). Lag values are indicated as coloured squares denoting months. Correlations were calculated locally in time for fragments of 25 months, a window size known to adequately trace ENSO effects [33].

Scale-Dependent Correlation Analysis (SDC [33,34]) was applied to examine correlations that are local in time between the different cholera strains and ENSO (Fig 5). Also, Multichannel Singular Spectrum Analysis (MCSSA [35,36]) was applied to decompose cholera variability into major (orthogonal) components accounting for the largest amount of its variance, and for time series reconstruction based on given components (with an application to epidemiology [37]). Here, reconstructions are specifically obtained by removing the trend and keeping all other components (Fig 2B), including seasonality and interannual variability (Fig 2B).

Climate variability in both space and time was decomposed via Principal Components Analysis (PCA) to identify, and discriminate among, main modes of variability in a particular variable, and then address associations of the dominant temporal modes (Principal Components, or PCs) with other variables (cholera levels [38]). The resulting empirical orthogonal functions (EOF) provide dominant spatial components over which temporal variation in the form of PCs can be examined (S2 Fig).

Finally, to assess the degree of severity of the climate anomalies that accompanied the historical epidemiological events, and to establish expected changes in the future under climate change, we analysed future temperature and precipitation projections in the relevant Bengal

region (Fig 6). We used the "business-as-usual" future scenarios RCP8.5 and SSP585 from the latest Coupled Model Intercomparison Projects, Phase 5 (CMIP5) and Phase 6 (CMIP6). We obtained monthly precipitation and surface temperature from 30 different climate models for each CMIP generation (see [39] for the full list of models). The future periods considered are

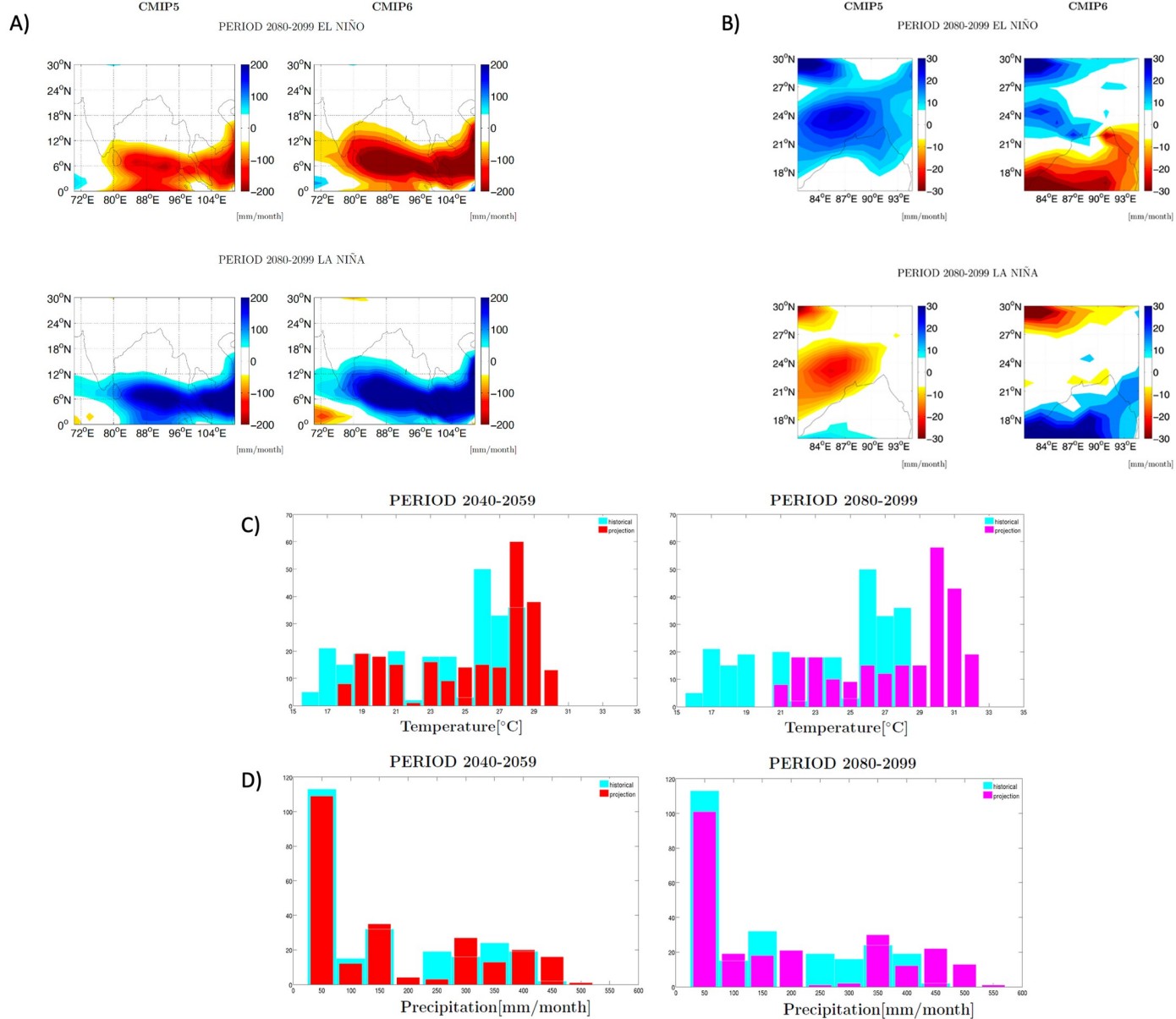

**Fig 6. Future climate projections for Bangladesh.** A) Differences in rainfall anomalies during A (top) El Niño or A (Bottom) La Niña events, expressed in [mm/month] between CMIP5 and CMIP6 multi-model simulations averaged over the region (87.5 to 92.5 deg. longitude and 20 to 27.5 deg. Latitude), corresponding to current Bangladesh. B) CMIP5 and CMIP6 projections for precipitation for the RCP8.5 and SSP585 scenarios as in A) but centered over a region covering former Bengal around present-day Bangladesh and parts of Assam and Meghalaya districts in India. C-D) Distribution changes for C) temperature and D) rainfall in future intervals (left: 2040–2059; right: 2080–2099) in CMIP6 projections. The distribution denotes the number of times a simulated value is obtained for each model. For each model output, the points considered were those only where the corresponding station data had more than 50% of rainfall measurements present. The data shown correspond to the multi-model mean of historical experiments (RCP8.5 and SSP585 scenarios in CMIP5 and CMIP6, respectively). Blue denotes the historical period whereas the future is indicated by both red (2040–2059) and pink (2080–2099). The list of climate and earth system models is given in ref. [39].

middle of the century (2040–2059) and end of the century (2080–2099) and the historical period is from 1980 to 1999. In Fig 6A and 6B we defined ENSO events in the usual way based on surface temperature anomalies in the Niño3.4 region[39] (5˚S– 5˚N, 170˚W– 120˚W). Anomalies are defined here with reference to a climatological mean computed separately over the historical and over the end of the century future period. El Niño and La Niña precipitation composites for December-January-February (DJF) are calculated in every model from CMIP5 and 6 ensembles by summing the precipitation anomalies occurring during each ENSO event. For Fig 6C and 6D we calculated histograms of precipitation and temperature ensemble means averaged over the region of Bangladesh (20˚N– 27˚N, 88˚E– 92˚E).

For readers not familiar with the previously described statistical methods we provide additional description in the S1 Supplementary Information and we guide interpretation of particular results in the text below and figure captions.

## Results

### Temporal patterns in historical cholera mortality

The 6th cholera pandemic commenced with a dramatic surge of cholera deaths in former British India, its endemic "homeland". In 1900 an unprecedented 700,000 deaths were reported from the provinces. In the Bombay Presidency, the low baseline rate from around 1 cholera death per 1000 increased nine-fold. Mortality was not above normal in Bengal, the most endemic province in 1900 (Fig 1), but a very steep rise in the case fatality was apparent from the harbor population in the capital Calcutta, suggestive of the introduction of a new strain (Fig 1). By 1905 cholera deaths in Calcutta rose sharply to over 180,000, two standard deviations above the mean, and the fraction of villages in Bengal with cholera mortality rose over the whole vast deltaic region (Fig 1) [40]. Also, the case fatality of the "floating population" rose again in 1905, before settling in 1912 to the normal rate of about 20% (Fig 1). Strikingly, the vast seasonal variation in cholera mortality between the 24 districts of Bengal vanished in 1905, with a marked development of synchrony and a shift of deaths to the fall (Figs 1 and S1B and S1C).

The synchrony of this change in mortality between districts in Bengal is shown in Fig 2A. The cross-wavelet power spectrum was computed between monthly cholera deaths in Dacca (used as reference) (Fig 2A) and individual districts along the Brahmaputra basin. Typically, the two coastal districts display a distinct seasonality [41]. But around 1905 all district pairs exhibited full synchronization. This spatio-temporal synchrony is evident for the annual cycle (12 months in the vertical scale, Fig 2A), indicating a temporary amplification of the seasonal cholera cycle.

Reconstructed components covering all time scales for cholera except the trend generated with Singular Spectrum Analysis (SSA [35,36]) for each time series in Bengal, lend further support to the anomalous synchronization across the region (Fig 2B). The less striking synchronization in 1893 (Fig 2A), and the increased mortality in 1927 in the absence of synchronization (Figs 1 and 2B) may reflect strain diversification with weaker consequences.

We interpret the spatial synchronization in this endemic region as a manifestation of a new strain of cholera for which the population lacked protective immunity. These changes in 1905 resemble those in the 1960s, when the *El Tor* strain replaced the "*Classical*" one on its march to global hegemony; with similarities extending to a delay in strain replacement, a shift of seasonal deaths to the fall period (as described below) and a raised case fatality rate. Data to investigate a shift of morbidity and mortality to an older age-group, signifying failure of acquired immunity, documented during *El Tor* and *O139* strain replacements were not available for Bengal. However available data by age and sex (1908–1919) for the Punjab province in NW part of the subcontinent show such a shift in 1911 (S1A Fig). Apart from the ratio between

cholera mortality in the population over and under 10 years of age increasing five-fold, an increase in the male-to-female ratio (>10 years) has been observed in non-endemic areas struck by a cholera [42], and is presumably related to a higher exposure of the male population.

## Climate conditions accompanying anomalous patterns of cholera mortality

The noticeable delay between the apparent introduction of a new virulent strain and its ability to outcompete the incumbent one, suggests other determinants may have been at play [43]. Weather is a prime potential candidate we investigate next. Fig 3A summarizes the anomalous seasonal patterns of mortality, rainfall and temperature in Dacca during the exceptional period of 1904–07 (calculated monthly, as the composite difference with regard to the long-term monthly average for 1893–1939). In contrast to the typical bimodal seasonal pattern of the disease with two peaks per year, the anomalous seasonality was characterized by a single, but extremely large, annual cholera peak (Figs 3A and S1B and S1C). Specifically, the spring peak was absent and the winter cholera peak grew in conjunction with decreasing rainfall after heavy and delayed monsoon rains in an environment where temperature is suitable for cholera proliferation. These conditions typically lasted until December when temperature decreased by more than 2.1 degrees. Anomalous high cholera incidence occurred during the three winters, particularly in 1904/05 and 1905/06 (S1C Fig), following anomalous monsoon rains, initially lower than normal for June (1904 to 1906), and extremely high values in August 1905 (*with more than 150mm excess rain*) and August 1906 (*over 75 mm excess rain*). These monsoon seasons were also preceded by anomalous spring conditions in both rainfall and temperatures. February 1905 experienced extreme low temperatures (Figs 3C and S1C), more than 3 standard deviations below the seasonal average, a situation that did not return to normal until late spring. In conjunction with this severe decrease in temperatures, a concomitant abnormal increase in rainfall occurred in March 1905 (Fig 3B, right) with absolute values more than double the typical average.

Comparison of the 1904–07 event with normal years for climate conditions (Figs 3A and S1B and S1C) highlight the delayed and more intense monsoon, with the rainfall maximum in August, and a long winter characterized by low temperatures that last until April. Among these months, anomalous higher than normal rainfall is again observed in February and March 1905. Exactly the same conditions are experienced over most of the country in 1905 (Fig 3B right and 3C right), with very cold anomalies already beginning the previous winter and with an intense and delayed monsoon. These same patterns occurred during the extreme cholera anomaly of 1893 (Fig 3B left and Fig 3C left).

Rainfall variability over the region for 1901–1939 exhibits a marked spatial structure as revealed by Principal Component Analysis (PCA) of the deseasonalized GPCC gridded reanalysis data (S2A Fig). The two main spatial components (the first and second Empirical Orthogonal Functions, EOFs) for rainfall display a NE-SW axis of variability identifying a dominant pattern either along the Ganges (NW-SE) or the Brahmaputra (NE-SW; S2A Fig) respectively. The first temporal PCs (PC1) corresponding to the main spatial mode (S2B Fig), indicates a high rainfall score over the whole country for 1905. Moreover, both PC1 and PC2 are significantly correlated with ENSO (*p<0.01*), consistent with a strong relationship between rainfall and ENSO known in the literature to be stable at least until late 1970s [44] (when an ocean regime shift occurred in the Pacific). The two first PCs of GPCC are very similar to those found in the station rainfall data (S2A Fig second row). Therefore, high scores in PC1 might also be related to large ENSO variability according to these analyses.

## ENSO teleconnections and cholera spread

The years of 1893 and 1905 correspond to the two major ENSO anomalies (in absolute terms) occurring in the interval of 1880–1940, namely a La Nina event in 1893 and an El Nino one in 1905 (Fig 4A). ENSO-related teleconnections in rainfall and temperatures have been associated in different parts of the world to the more recent temporal patterns of cholera dynamics [45–48]. For the specific climate anomalies in Bangladesh, both the 1893 LN and the 1905 EN led to similar teleconnection patterns (e.g. massive flooding and extreme low temperatures). EN and LN events may in some regions and for some events, produce similar climatic anomalies. For instance, among the eight El Niño events between 1981 and 2021, seven of them corresponded to above-normal Bangladesh Summer Monsoon Rainfall (BSMR). However, during the 11 La Niña events, the relationship was more varied, with above-normal BSMR occurring in seven instances [49]. These findings indicate an asymmetric relationship between BSMR and ENSO.

We can complement these findings by focusing here on cholera cases during the global expansion of the 7th pandemic and to ENSO's regional associated teleconnections in Asia (1963–64), Africa (1991–92; 1997 and 2002) and the Americas (1991–92). Fig 4B provides a summary of these temporal and spatial links with ENSO events and places the 1893 and 1905 ENSO events in context [50].

For comparison, we also examined the contribution of ENSO to cholera's exacerbation at specific times of strain replacement in present-day Bangladesh around the second half of the XXth century (Fig 5). To investigate ENSO's influence on regional climate, Scale-Dependent Correlation Analysis (SDC) [51] was applied between global sea-surface temperature (SST) anomalies and ENSO (i.e. Niño 3.4 index). The resulting maps show moving-window correlations calculated between the Niño 3.4 index and the SST series at each grid point of a 2.5 degrees' resolution of a global database (Fig 5A). These maps illustrate the anomalous oceanic configurations occurring during the three El Niños (EN), including a strong (1987) and very strong (1982 and 1997) events known to have operated differently over the Indian subcontinent [37,40,52] (Fig 5A). Specifically, in our analysis the identified areas of significant correlations in global ocean SSTs are distinct among the three EN events. This is consistent with changes in ocean SST values with differential impacts over the Indian subcontinent.

These SST changes occur at times that are associated with cholera events as shown in Fig 5B, where A panels show SST anomalies linked to individual EN events, whereas B panels demonstrate differential impacts on cholera strains. This can be seen in the SDC analysis here applied to correlations between ENSO (Niño3.4 index) and total cholera cases, individually for each of the three different strains in Bangladesh (Fig 5B for *Classical*, *El Tor* and *O139*, *a variant of the El Tor strain*). In all cases, ENSO led to cholera by between 7 to 12 months, a delay consistent with previous studies [45,53]. Strikingly, local monthly correlation maxima clustered around the years when cholera experienced the strains' replacements, as well as the large increases in cases for each of the three strains. In particular, these clusters are clearly found for the *Classical* strain following the very strong El Niño event of 1982, for *El Tor* replacing *Classical* in 1986–88, and for *O139* overtaking *El Tor* from1994 to 1997.

## Global climate simulations to evaluate the rainfall anomalies in the past and future

Finally, to put in context the degree of severity of the 1905 positive rainfall anomalies during the delayed monsoons, and to consider the expected changes in the future under the harsher evolution of greenhouse gas emissions (i.e., RCP8.5 and SSP585 scenarios in CMIP5 and CMIP6, respectively), the results of multi-model climate simulations of temperature and precipitation [39] for the Bangladesh region are shown in Fig 6. A comparison between CMIP5

and CMIP6 simulations for the end-of-century (2080–2099) period is also made to better highlight the different impact of ENSO that they project on local regional rainfall (Fig 6A) and in Bangladesh (Fig 6B). Model projections indicate a stronger and more coherent impact of ENSO on precipitation in Bangladesh in CMIP5 than in CMIP6 (Fig 6A and 6B), but they generally agree on wet conditions during El Niño and dry/neutral conditions during La Niña. Fig 6C and 6D show the temperature and precipitation histograms for Bangladesh for the middle and the end of the century periods as projected by CMIP6 models. A major shift towards warmer temperatures is visible in Fig 6C, especially for the end-century period when an increase of up to ~4˚C is observed for Bangladesh. A positive shift of the precipitation distribution in the future as compared to the historical period is also seen in Fig 6D. The most extreme amounts of monthly precipitation (450–550mm/month) are also projected to occur more often in the future.

## Discussion and conclusions

The pandemic excursions of cholera remain among the least understood phenomena in infectious disease research. The anomalous complete synchronization event described here for Bengal between 1904–1907 suggests a major external force was at play on the otherwise spatially differentiated cholera dynamics of this region. One possible explanation for such a synchronous and extreme cholera event is climate anomalies acting over a large geographical region (i.e. Moran effect [54,55]). Another explanation is the emergence of a novel strain. Acting on top of a secular trend in mortality (Fig 1) explained by improvements in treatment and sanitary conditions [24], the introduction and full establishment of a new and possibly more virulent strain would result in higher mortality at the regional scale. As documented for the most recent succession of strains [41,42], an increased mortality due to strain emergence would be accompanied by a relative shift in mortality to older age groups. This shift is explained by a lack of immunity in age groups with protection against the previously dominating strain. For the historical data from former British India, we relied here on these known features of changing strains to retrospectively assess the evidence for a strain shift during the 6th pandemic (1899–1927). The unique spatial resolution of these records allowed us to examine cholera's synchronization over an unprecedented large region, across districts, and in the context of a long temporal record.

The hypothesis that strain variation has been responsible for the recurrent changes in apparent virulence associated with pandemics [56] is difficult to test, as strain identification only commenced after isolation of the bacterium during the 5th pandemic. By using other epidemiological characteristics of the appearance of new strains in recent decades, we presented evidence consistent with a novel strain in the initial stage of the 6th pandemic. A gap exists, however, between the first recognised outbreaks of the 6th pandemic around 1900 (Calcutta and Bombay) and the highest and synchronous all India cholera mortality around 1905. This delay does not necessarily contradict that these two events, first appearance and emergence, are related. A new strain might have made its first appearance in 1899–1900, becoming fully established and replacing the previous strain a few years later. Such a sequence of events was indeed observed for the replacement of the *Classical* strain by *El Tor* in the same area at the beginning of the 7th pandemic. *El Tor* accounted for a small proportion of cases in a Dacca Hospital between 1968 and 1972, and it completely replaced *Classical* in 1973 for a number of years [57]. After some time, there was a reoccurrence of the *Classical* strain and the apparent spatial differentiation of both strains in Bangladesh [58]. Similarly, *El Tor's* genetic shifts occurred in 1992 in China related to the replacement of the prototype strain from 2002 to 2010 [10]. Altered variants of atypical *El Tor* biotypes were also identified in India and Mozambique

and completely replaced former cholera isolates in Thailand [11], Vietnam [59] and Angola [12] around 1991.

For all those years and places, El Niño conditions were documented in previous studies, consistent with the notion that altered climate variability favors cholera epidemics. Recent large-scale disease events took place regionally during periods of time that encompassed strong El Niño (EN) years, as noted for South America during 1991–92 [9] and for the large outbreaks in Africa after 1997/98 and 2002/03. Disease synchronization among regions affected by strong EN events has also been documented for the latter outbreaks [46].

There was no shift in the seasonality of cholera after 1905, a pattern observed later when *El Tor* replaced the *Classical* strain [57] with a well-documented winter peak shift from December to mid-October, and also when *O139* appeared in the 1990s. Differences in seasonality between strains, most likely reflect different ecological requirements between strains [60,61] that do not necessarily accompany antigenic and virulence changes. We do document a higher adult male mortality in the Punjab a few years after the 1904–07 event, a characteristic of the more recent replacements. Similar changes in epidemiology were reported in former strain shifts (i.e. the Danish cholera outbreak of the 1850s [16]). These include a change in virulence also reflected in the case-fatality rate (for *O139*) and a shift of the disease burden to an older population (*El Tor* and *O139*), associated with the lack of specific immunity to the new strain.

Several of our results support a role of climate acting as a major driver of the 1904–07 anomalous cholera episode, which would have facilitated the establishment of the novel strain. The atypical suppression of the cholera spring peak during the 1904–07 abnormal cholera interval was shown to be related to anomalous cold conditions and high precipitation. In addition, higher mortality in the dominant winter peaks would have been facilitated by the anomalous monsoon season of the previous summer. Thus, different drivers appear to underlie the two cholera peaks in a normal year, consistent with studies of environmental factors determining the seasonality of the disease. Very low temperatures in late winter would have helped in sharply ending the winter cholera peak. Similarly, low temperatures continuing through mid-spring (of the order of two to three degrees lower than normal) and concurrent high relative rainfall (an excess of more than 1,000 mm) might have led to the full suppression of the spring cholera peak that normally follows. Temperature during these months has been shown to affect the intensity of the spring peak [24]. Both effects would have generated a larger pool of susceptible individuals or/and lower levels of temporary acquired immunity, contributing to a large winter outbreak in the transmission season that followed the monsoons. Extremely heavy rains in the monsoon season may also have played an important role by disrupting sanitation systems and promoting with a delay the proliferation of bacteria in the environment due for example to increased nutrients. Evidence for a dual role of rainfall in the seasonal cycle of cholera with both a negative and a positive effect at different lags has been presented for historical cholera in endemic regions of Madras [60] and former Bengal [61]. Anomalous rainfall conditions can also affect harvest levels, enhancing population malnutrition and promoting famine and disease [62].

In relation to the described regional anomalies in rainfall and temperature, several of our results support a role of ENSO during the 1904–07 event, consistent with studies of more recent cholera and climate variability [9,40–44]. In particular, the strong quasi-quadrennial component of the mortality (with a 4-yr periodicity) in the 1899–1905 interval observed for cholera (Fig 2A) is consistent with the strong coupling of ENSO to monsoon dynamics, as indicated by other studies [63,64].

In summary, our results indicate that genetic shifts in the dominant *Classical* strain together with facilitation by climate underlie the establishment and expansion of the 6th pandemics. These concomitant effects have implications for the potential future emergence of other

pathogenic strains, as they underscore the role of environmental conditions in generating windows of opportunity for the colonization by novel variants. Recent observed changes in cholera strains already include the shift towards *El Tor* dominance and the several failed attempts by *O139* to replace *El Tor*. Interestingly, strong El Niño conditions co-occurred with successful strain changes in the last decades of the 20th century [33]. For the earlier 6[th] pandemic studied here, during the two EN years of 1899 and 1905 most inter-hemispheric and global indicators of global variability manifested an intense peak in activity [64].

In view of this clear connection between strong EN years and cholera strain changes, it is important to assess the projected ENSO variations in the future, especially under climate change. As shown by Petrova et al. in [39], in both future periods considered here—mid-century and end-century, the CMIP6 ensemble of climate models projects more ENSO events as compared to the historical period, and this result is more robust in CMIP6 than in CMIP5 (i.e. more models project an increase in CMIP6). This implies a stronger climatic impact of ENSO in the CMIP6 model ensemble. The CMIP6 simulations also suggest more warm events relative to cold events as compared to the older CMIP5 generation [39]. CMIP5 and CMIP6 both agree on the future increase of extreme ENSO events [65–69], and some of these studies suggest that their number will double in the future compared to the historical period. This also means that locations whose rainfall is impacted by ENSO will experience much more often extreme positive or negative precipitation events. Noticeably, we find stronger impact of ENSO on future precipitation in Bangladesh in CMIP5 than in CMIP6 (Fig 6A and 6B). However, the CMIP6 simulations suggest an increasing frequency and intensity of extreme positive precipitation events in the region of Bangladesh (Fig 6D). The majority of CMIP6 models also foresee an increase of the ENSO amplitude by the end of the century under the extreme emissions scenario [70], and it is important to note that this is not the case in CMIP5. These results collectively point to a higher probability for the emergence of cholera pathogens until the end of the century.

Finally, we specifically demonstrated the extreme nature of climate conditions around 1905 by placing these in the context of monsoon rainfall distributions generated by climate models. Those conditions proved extremely rare for the past and also for the future under a climate change scenario. This should not be interpreted as indicating that similar magnitudes of anomalies would be required for pathogen emergence. More likely, climate conditions that enhance transmission during particular windows of time and several consecutive years can act to facilitate emergence in cholera, but also in other pathogens that are water-borne and vector-borne and therefore, closely connected to the environment. The increased capacity to monitor genetic changes of pathogens should be coupled to advances in climate studies of infectious diseases including climate modelling and prediction, to provide warnings about potential emergence. The finding of a possible delay between initial detection and actual emergence indicates that the relevance of climate conditions can extend beyond the former. For cholera, the shift of the distribution of monsoon rains over Bangladesh towards higher values suggests more frequent conditions for transmission and emergence. The synergy of strain variation and anomalous environmental conditions arising more often, provides a warning of continued and enhanced opportunities for emergence of the disease given the strong climate change signals in its homeland and beyond.

Because of its historical nature, a limitation of our study is necessarily a reliance on the statistical analyses of temporal and spatio-temporal patterns, including the unusual large-scale synchronization and its association with climate conditions. More than one century later, at a time when strain genetic variation can be easily accessed, the interplay of pathogen evolution and anomalies in climate at different temporal scales should become an integral part of understanding and predicting the population dynamics of infectious diseases that are climate-sensitive.

## Supporting information

**S1 Supplementary Information. Contains explanations about datasets, methods and supplementary figures.**
(DOCX)

**S1 Fig.** Relationships between rainfall ($10^{-1}$mm/month) and temperature (deg. ˚C/day) in Dacca for the reference period of 1893–1935, together with anomalies in cholera mortality: A) distribution of monthly rainfall (NEIR) and temperatures, with means indicated by large circles (J—January, F—February, Mr—March, Ap—April, Au—August, S—September, O—October, N—November, and D—December); B) distribution of total occurrences of cholera mortality anomalies during the monsoons (May to September) and in winter (October to April); C) evolution of cholera mortality, rainfall and temperature during the anomalous 1904–07 event (grey background stripes correspond to the monsoon period).
(TIF)

**S2 Fig.** A) Principal Component Analysis (PCA) of the de-seasonalized GPCC rainfall reanalysis [25] over Bangladesh for the interval 1901–1940. Years before 1901 were not available. Empirical Orthogonal Functions (EOF) are significant at the $p < 0.01$ level. Bottom row depicts the results of a similar PCA but applied on ground station data. B) Temporal PCs (t-PC) for the EOF components in A) with red years denoting large cholera anomalies in Bengal (*note that the sign is arbitrary in PCA*).
(TIF)

## Acknowledgments

We thank the Indian Institute for Tropical Meteorology at Pune (http://www.tropmet.res.in) for supplying meteorological data.

## Author Contributions

**Conceptualization:** Xavier Rodó, Menno Jan Bouma, Mercedes Pascual.

**Data curation:** Xavier Rodó, Menno Jan Bouma, Miquel-Àngel Rodríguez-Arias, Manojit Roy, Pau De Yebra, Desislava Petrova, Markel García-Díez, Mercedes Pascual.

**Formal analysis:** Xavier Rodó, Miquel-Àngel Rodríguez-Arias, Manojit Roy, Pau De Yebra, Desislava Petrova, Markel García-Díez, Mercedes Pascual.

**Funding acquisition:** Xavier Rodó, Mercedes Pascual.

**Investigation:** Xavier Rodó, Menno Jan Bouma, Miquel-Àngel Rodríguez-Arias, Manojit Roy, Pau De Yebra, Desislava Petrova, Markel García-Díez, Mercedes Pascual.

**Methodology:** Xavier Rodó, Menno Jan Bouma, Miquel-Àngel Rodríguez-Arias, Pau De Yebra, Mercedes Pascual.

**Project administration:** Xavier Rodó.

**Software:** Xavier Rodó.

**Supervision:** Xavier Rodó, Menno Jan Bouma, Mercedes Pascual.

**Validation:** Xavier Rodó, Mercedes Pascual.

**Visualization:** Xavier Rodó, Desislava Petrova.

**Writing – original draft:** Xavier Rodó, Mercedes Pascual.

**Writing – review & editing:** Xavier Rodó, Menno Jan Bouma, Pau De Yebra, Desislava Petrova, Mercedes Pascual.

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
