## [Decision Letter · Decision Letter 0]

12 Apr 2024

Dear Dr. Rodo,

Thank you very much for submitting your manuscript "Strain variation and anomalous climate synergistically influence cholera pandemics" for consideration at PLOS Neglected Tropical Diseases. As with all papers reviewed by the journal, your manuscript was reviewed by members of the editorial board and by several independent reviewers. The reviewers appreciated the attention to an important topic. Based on the reviews, we are likely to accept this manuscript for publication, providing that you modify the manuscript according to the review recommendations. 

Your manuscript has been reviewed by two experts in the field. Generally they were positive about the study and it should be straightforward to address their concerns. Please respond point by point to each reviewer comment and included with your revised manuscript.

Sincerely,

Jeffrey H Withey

Academic Editor

Elsio Wunder Jr

Section Editor

Your manuscript has been reviewed by two experts in the field. Generally they were positive about the study and it should be straightforward to address their concerns. Please respond point by point to each reviewer comment and included with your revised manuscript.

Reviewer's Responses to Questions

**Key Review Criteria Required for Acceptance?**

**Methods**

-Are the objectives of the study clearly articulated with a clear testable hypothesis stated?

-Is the study design appropriate to address the stated objectives?

-Is the population clearly described and appropriate for the hypothesis being tested?

-Is the sample size sufficient to ensure adequate power to address the hypothesis being tested?

-Were correct statistical analysis used to support conclusions?

-Are there concerns about ethical or regulatory requirements being met?

Reviewer #1: -Are the objectives of the study clearly articulated with a clear testable hypothesis stated? Not quite. See my full review.

-Is the study design appropriate to address the stated objectives? Probably yes.

-Is the population clearly described and appropriate for the hypothesis being tested? Yes.

-Is the sample size sufficient to ensure adequate power to address the hypothesis being tested? Yes.

-Were correct statistical analysis used to support conclusions? Probably, but I have some questions. See my full review.

-Are there concerns about ethical or regulatory requirements being met? No.

Reviewer #2: (No Response)

**Results**

-Does the analysis presented match the analysis plan?

-Are the results clearly and completely presented?

-Are the figures (Tables, Images) of sufficient quality for clarity?

Reviewer #1: -Does the analysis presented match the analysis plan? Mostly, but see the concerns in my full review.

-Are the results clearly and completely presented? Sort of.

-Are the figures (Tables, Images) of sufficient quality for clarity? Not really.

Reviewer #2: (No Response)

**Conclusions**

-Are the conclusions supported by the data presented?

-Are the limitations of analysis clearly described?

-Do the authors discuss how these data can be helpful to advance our understanding of the topic under study?

-Is public health relevance addressed?

Reviewer #1: -Are the conclusions supported by the data presented? Mostly, yes.

-Are the limitations of analysis clearly described? Not fully. See my full review.

-Do the authors discuss how these data can be helpful to advance our understanding of the topic under study? Yes - future pandemics.

-Is public health relevance addressed? Yes.

Reviewer #2: (No Response)

**Editorial and Data Presentation Modifications?**

Reviewer #1: See my full review.

Reviewer #2: (No Response)

**Summary and General Comments**

Reviewer #1: The authors examine the role of climate (temperature, precipitation, and El Niño) in the cholera pandemic of 1899-1923) in former British India (Bengal, or present-day Bangladesh), and generate climate-based projections. The authors conclude that rainfall anomalies toward the beginning of the outbreak were extreme (statistically) and played a role, and that increasing variability and frequency of such extremes is of concern for future pandemics.

For context, I’m a physical climate scientist with relatively little experience with public health, epidemiology, etc. I’ve collaborated on some climate/health studies, but my contributions have largely been in the climate sphere (observations and global climate modeling). My comments mostly focus on those aspects of this study.

My overall impression is that this is a nice study worthy of publication in PLOS NTD after some revisions. It’s clear to me that the authors are disease folks who are doing their best with climate observations and models. I mean that in a nice way. I give the authors an A on the novelty and probably everything else, but a B at best with the climate analysis, and I don’t think they’re going to get to an A without collaboration with a climate specialist. To be blunt, most of the climate stuff here would be torn apart in a climate journal, but that’s not the point of interdisciplinary studies like this in PLOS NTD. What’s important is that I don’t think anything with the climate aspect of this study is concerning enough to me that it undermines the overall conclusions. I actually buy the conclusions.

INTRODUCTION

I enjoyed the introduction. I found it easy to follow and well referenced. Just one minor thing—on page 5, “interaction” is probably not the right word, unless you’re suggesting the pandemic feeds back onto climate.

DATA AND METHODS

The cholera dataset seems like an amazing resource. The data are organized monthly by villages, so the spatiotemporal sampling and resolution seems appropriate for the analysis objectives. Reliable climate data from this period (turn of the 20th century) are challenging to come by, but due in part to the colonial enterprise, relatively good records from this region were kept at the time. I have a few comments about the climate data.

The datasets used seem appropriate, but note that SST is not really “from” the 20th Century Reanalysis. That is an atmospheric reanalysis/data assimilation product and SSTs are prescribed rom another source. That sources is called SODA. Please review the information on this page under the Description tab: https://www.psl.noaa.gov/data/gridded/data.20thC_ReanV3.html

Which version of the NOAA 20th Century Reanalysis is used? There are three at this point.

You refer to the Global Precipitation Climatology Product, but then use the acronym GPCC. The problem is that there actually are two different precip datasets: GPCP and GPCC. I believe you are using GPCC but please correct the acronym to eliminate the ambiguity.

The statistical techniques seem standard and appropriate, including the use of EOF analysis with the climate data.

I have an issue with the climate models, though. The scientific community has had free and open access to CMIP6 for many years now. Without a specific justification for using CMIP5 rather than CMIP6, I cannot say I approve of this. … However, I note that in the SI (and only in the SI), it is stated that CMIP5 and CMIP6 models are used. This needs to be harmonized with what is presented in the main text. I later see that Fig. 5 includes both generations of models, and both are discussed in the Discussion and conclusions section. I guess the methods section in the main text just needs to be updated to indicate that CMIP6 models are also included.

The authors say that one thousand multi-model simulations based on CMIP5 were averaged. I’d like to understand where the number 1,000 comes from. According to the SI, the authors used 30 models from CMIP5 and 30 models from CMIP6. That implies an average of 16.7 ensemble members from each model are being used. I do not see discussion of ensemble members, so it’s not clear where this is coming from. From the description in SI, it almost sounds like the authors mean members of a composite (compositing on ENSO phases), which is not the same as ensemble runs (where the initial conditions are varied) from climate models.

RESULTS

It’s not clear what is plotted in Figure 3A. The caption does not help. Difference between the average of the 1904-08 event and the seasonal cycles of … over what time period? What is being differenced, exactly?

What do the authors mean by a 21 degree temperature drop? I don’t see anything like that on any of the figures. Is the red line on 3A scaled by 0.1 or something?

Are the EOF results shown in S2 conducted on annual mean precip, seasonal mean precip, or monthly? If monthly, I really hope the authors are removing the climatological seasonal cycle first, otherwise the first mode will of course be the annual cycle. These details need to be provided in the methods section(s), if not the captions as well.

Please don’t show the 3rd and 4th mode. They are obviously statistical noise and are (thankfully) not analyzed.

Page 11: Just a friendly writing suggestion: sentences like “highly anomalous cholera years are indicated as red dots” belong in captions, not in main text.

The global correlation maps between SST and NINO3.4 (Fig. S3A) are kind of bizarre, I have to say. There’s nothing new there, and it’s not clear how those connect to cholera (S3B) and S3B is extremely hard to read. I cannot make out the years on the x-axis. It’s possible there is something really important here, especially with the supposed result that cholera lags ENSO by 7-12 months, but I definitely don’t see it from reading the top paragraph on page 13 and looking at Fig. S3.

The actual role for climate models (CMIP5/6) in this paper is very minor. It boils down to a pretty poorly done figure tacked on at the end of the paper (Fig. 5). The color scales in the maps are extremely saturated and no indication of statistical significance is given. Also, annual? Seasonal? All months? Note that if annual, it’s not a good idea to composite annual mean fields onto ENSO since ENSO peaks at the breakpoint between calendar years.

DISCUSSION AND CONCLUSIONS

Nice summary and connection to climate literature about the future of ENSO. No concerns here. I don’t detect exaggeration of the results or their significance.

Reviewer #2: (No Response)

PLOS authors have the option to publish the peer review history of their article (what does this mean?). If published, this will include your full peer review and any attached files.

Reviewer #1: No

Reviewer #2: No

Figure Files:

Data Requirements:

Reproducibility:

References

---

## [Editor Report · Decision Letter 1]

6 Jun 2024

Dear Dr. Rodo,

We are pleased to inform you that your manuscript 'Strain variation and anomalous climate synergistically influence cholera pandemics' has been provisionally accepted for publication in PLOS Neglected Tropical Diseases.

Best regards,

Jeffrey H Withey

Academic Editor

Elsio Wunder Jr

Section Editor

---

## [Editor Report · Acceptance letter]

25 Jun 2024

Dear Dr. Rodó,

We are delighted to inform you that your manuscript, "Strain variation and anomalous climate synergistically influence cholera pandemics," has been formally accepted for publication in PLOS Neglected Tropical Diseases.

Best regards,

Shaden Kamhawi

co-Editor-in-Chief

Paul Brindley

co-Editor-in-Chief
